# Physiochemical Quality, Microbial Diversity, and Volatile Components of Monascus-Fermented Hairtail Surimi

**DOI:** 10.3390/foods12152891

**Published:** 2023-07-29

**Authors:** Yanpo Li, Shuyi You, Lujie Cheng, Hongliang Zeng, Baodong Zheng, Yi Zhang

**Affiliations:** 1Engineering Research Center of Fujian-Taiwan Special Marine Food Processing and Nutrition, Ministry of Education, Fuzhou 350002, China; liyanpo@wzvcst.edu.cn (Y.L.); ysyfst@fafu.edu.cn (S.Y.); 1210919010@fafu.edu.cn (L.C.); zhlfst@fafu.edu.cn (H.Z.); zbdfst@fafu.edu.cn (B.Z.); 2College of Food Science, Fujian Agriculture and Forestry University, Fuzhou 350002, China; 3Department of Agriculture and Biotechnology, Wenzhou Vocational College of Science and Technology, Wenzhou 325000, China

**Keywords:** Monascus-fermented hairtail surimi (MFHS), microbial diversity, fermentation, volatile compounds, correlation

## Abstract

In order to study the effects and mechanism of Monascus on the quality of hairtail surimi, high-throughput sequencing technology, headspace solid-phase microextraction-gas chromatography-mass spectrometry (HS-SPME-GC/MS), and electronic nose techniques were used to investigate the changes in the quality, microbial diversity, and volatile flavor compounds of Monascus-fermented hairtail surimi (MFHS) during fermentation. The results showed that the total volatile basic nitrogen (TVB-N) index of hairtail surimi fermented by Monascus for 0–5 h met the requirements of the national standard. Among them, the 1 h group showed the best gel quality, which detected a total of 138 volatile substances, including 20 alcohols, 7 aldehydes, 12 olefins, 4 phenols, 12 alkanes, 8 ketones, 15 esters, 6 acids, 16 benzenes, 4 ethers, and 8 amines, as well as 26 other compounds. In addition, the dominant fungal microorganisms in the fermentation process of MFHS were identified, and a Spearman correlation analysis showed that 16 fungal microorganisms were significantly correlated with the decrease in fishy odor substances in the fermented fish and that 8 fungal microorganisms were significantly correlated with the increase in aromatic substances after fermentation. In short, Monascus fermentation can eliminate and reduce the fishy odor substances in hairtail fish, increase and improve the aromatic flavor, and improve the quality of hairtail surimi gel. These findings are helpful for revealing the mechanism of the quality formation of fermented surimi and provide guidance for the screening of starter culture in the future.

## 1. Introduction

The global annual per capita consumption of aquatic foods increases year by year, and it grew from 9.9 kg in the 1960s to 20.5 kg in 2019. In China, the per capita consumption grew from 4.2 kg in 1961 to 40.1 kg in 2019 [1]. The textural features, nutritional benefits, and easier consumable convenience has driven surimi’s demand [2]. Surimi fish balls, sausages, and tofu have become daily food [3]. The surimi processing technique originated in the 1960s, when cold-water whitefish such as Alaska pollock were chosen as the major raw material for surimi manufacturing due to their superior gel characteristics and nutrition. As the market for surimi and surimi products blooms, a range of marine lean fish are being utilized as raw materials for surimi, including American arrow plaice, pacific cod, threadfish, and bigeye tuna [4]. Unfortunately, the fishing industry and the production of surimi are being severely hampered by overfishing, marine fish supplies depletion, and the availability of high-quality surimi sources reduction [5].

With a catch of 914,469 tons in 2021, China continues to come first in hairtail production through marine capture; given its low price and small body size, small hairtail surimi has become the primary raw material for surimi products. However, the gel performance of surimi products is significantly lower than that of other high-quality white meat fish because of the high-fat content, relatively low salt-soluble protein content, and fishy smell [6,7]. As a result, a certain amount of transglutaminase (TGase), food gum, phosphate, and other food additives must be added to enhance the gel quality and flavor of the product, which not only drives up the price of the item, but also poses some risks for food safety. Therefore, improving the quality of hairtail surimi is a key factor in effectively alleviating the scarcity of high-quality surimi raw materials, enhancing the added value of the surimi industry, and promoting the sustainable development of China’s marine fishing industry.

The common methods for improving surimi gel include two-stage heated gel [8], microwave-heated gel [9,10], ultra-high pressure gel [11,12,13,14,15], the addition of glutaminase [16,17], protease inhibitors [18,19], acidogenic gel [20,21,22,23,24], and biological fermentation gel [15]. One of the traditional Chinese food processing techniques is the use of fermentation to enhance product quality and storage life. Products such as wine lees hairtail and other food items may be enhanced through fermentation by producing a particular taste and by preventing the growth of dangerous bacteria while being stored. This improves the product performance and provides specific healthcare effects in addition to rich nutritional content and a delicious fermentation flavor [25,26].

Naturally fermented fish products contain lactic acid bacteria, micrococcus, mold, yeast, and other microorganisms in large quantities. Microbially fermented aquatic goods can enhance product safety and flavor [27,28]. Currently, *Lactobacillus* is utilized more frequently for anaerobic or micro-oxygenated fermentation. The production of lactic acid can lower the pH of the system, induce the acid-induced gel of myofibrillar protein, and inhibit the growth of pathogenic bacteria such as *Escherichia coli* and *Staphylococcus* aureus, thus improving food safety. Metabolites such as small-molecule peptides and free amino acids that are produced during biological fermentation can impart special flavor and color to surimi gel products as well as improve the nutritional value of surimi products. Xu et al. [29] used *Pediococcus pentosaceus* to ferment silver carp sausage and found that the higher temperature (37 °C) fermentation group inhibited the growth of *Pseudomonas*, *Micrococcaceae*, and *Enterobacteriaceae* but that the total amount of TVB-N and biogenic amines increased in the sausage. In addition, the fermented products exhibited the greatest consumer preference and best texture in the range of 23–30 °C. Nie et al. [30] inoculated *Lactobacillus plantarum* and *Fasciococcus pentose* in grass carp intestines to research the effects of fermentation on protein composition and structure, free amino acids, α-amino nitrogen, and TCA-dissolving peptides in sausages. The results showed that during fermentation, the sarcoplasmic protein and myofibrillar protein in the two grass carp sausages were significantly degraded, and the content of free amino acids, α-amino nitrogen, and TCA-dissolving peptides increased. Zhao et al. [31] systematically studied the dynamic changes of physicochemical properties, volatile compounds, and microbial community during the fermentation process of naturally fermented tilapia sausage; the authors concluded that Lactococcus, Staphylococcus, and Enterococcus played a key role in the formation of physicochemical and flavor characteristics. However, the accumulation of biogenic amines may result from the metabolism of Enterococcus, Enterobacter, and Citrobacter.

Multi-species compound fermentation is also used in surimi fermentation, such as adding yeast and mold to improve the flavor of surimi and enhance organoleptic quality [32,33,34]; these starter cultures play an important role in changing flavor, shortening fermentation time, extending shelf life, promoting color formation, and enhancing product safety. The addition of 1.5% *Monascus* can replace nitrite to increase the color of tuna-cured fish, which results in a better appearance, flavor, and taste [35].

Conventional Monascus fermentation technology exhibits the characteristics of complex microbial ecology, susceptibility to mixed bacteria contamination, and a fermentation process that is challenging to regulate. Thus, this paper mainly examines the application of Monascus fermentation to hairtail surimi raw materials and its impact on the quality of hairtail surimi. In addition, we study the mechanism from the perspective of the physical and chemical properties and the succession of fermentation microbial microbiota, as well as the established MFHS quality control technology. This study provides an important contribution for developing the strains that are appropriate for MFHS and for establishing uniform, standardized industrial production technologies for hairtail surimi.

## 2. Materials and Methods

### 2.1. Sample Preparation

Hairtail surimi were provided by Zhoushan Base of Haixin Group. Method of MFHS: Thawed hairtail surimi were brought to surface melting, chopped (FP3010, Braun/Braun, Taunus, Germany) for 2 min, had Monascus added to them (purchased from Chengjiu Monascus Co., Ltd., Gutian County, Ningde, China), and were crushed through an 80-mesh sieve. The preliminary test showed that a 0.1% added amount of fermented hairtail surimi had the best overall quality. The mixture was then well-mixed, loaded into sterile bags, and fermented in a water bath at 25 °C. Samples of Monascus-fermented hairtail surimi (MFHS) from each batch were collected at 0, 1, 3, 5, and 7 h and stored at −80 °C for a subsequent analysis. As a comparison, surimi without Monascus was used as a control.

Method to prepare the MFHS gel: The MFHS was chopped for 2 min, mixed with 2% NaCl for 5 min, and finally poured into a 42 mm plastic casing with the opening being tied in sections. A two-stage heat treatment was used to form the gel (30 min in a water bath at 40 °C and 30 min in a water bath at 90 °C), which was then cooled in ice water for 30 min and refrigerated at 4 °C for 12 h before the analysis.

### 2.2. Total Volatile Base Nitrogen (T-VBN)

T-VBN content in hairtail surimi samples was measured using the Kjeldahl method. We took 5 g samples that were homogenized according to the ratio of surimi:water = 2:1 and placed them in a distillation tube; we added 75 mL of water, shook the samples, and then immersed them for 30 min. We added 1 g of magnesium oxide into the distillation tube containing the sample, immediately connected it to the distillation machine, performed steam distillation for 5 min, distilled the liquid with 20 g/L of boric acid absorption, and then titrated the boric acid solution with 0.01 M of HCL standard titration solution, using methyl red and methylene blue as mixed indicators. The TVB-N values were determined using a K1100 automatic Kjeldmann nitrogen analyzer (Hanon, Jinan, China). The result uses Equation (1) to calculate the TVB-N value.
TVB-N (mg/100 g) = ((V_1_ − V_2_) × c × 14)/m × 100(1)
where V_1_ is the 1 is the titration volume for the tested sample (mL); V_2_ is the titration volume of blank (mL); c is the actual concentration of HCl (mol/L); m is the weight of the sample (g). The experimental results are expressed as the arithmetic mean of the results of three independent determinations that were obtained under reproducible conditions, and three significant digits were retained for the results.

### 2.3. Texture Profile Analysis (TPA)

The surimi gel was cut into a 25 mm high cylinder; a P/5 probe was selected, and the TPA mode was used. The speed was 60 mm/s during testing, and the downward pressure form variable was set to 50%. The average result was calculated after all experiments were performed nine times. The texture data of the surimi gels included the hardness, adhesiveness, cohesiveness, springiness, gumminess, chewiness, and resilience.

### 2.4. Water Holding Capacity (WHC)

According to the procedure described by Luo et al. [36], the hairtail surimi gel was cut into tiny pieces and accurately weighed to about 5 g (W1) before being placed into two layers of filter paper, centrifuged at 3750× *g* for 15 min at 4 °C, and finally weighed again (W2). We calculated the holding capacity of the hairtail surimi gel according to Equation (2).
WHC(%) = (W1 − W2)/W2 × 100(2)

### 2.5. Scanning Electron Microscope (SEM)

The method used by Wang et al. [37] was followed with a minor modification. The surimi gels were cut into 5 mm × 5 mm × 1 mm pieces, fixed in 2.5% glutaraldehyde solution for 24 h at 4 °C, rinsed several times with 0.1 mol/L of phosphate buffer (pH 7.2), and dehydrated with gradient ethanol solution (volume fraction 50%, 60%, 70%, 80%, 100%). After vacuum freeze-drying, gold was sprayed using a vacuum ion sputter while using a Nova NanoSEM model 230 scanning electron microscope (FEI CZECH REPUBLIC S.R.O Czech Republic, Brno-Černovice, Czech Republic) in SEM mode at a voltage of 10 kV, which was performed for the microstructure analysis of hairtail surimi gels.

### 2.6. Low-Field-Nuclear Magnetic Resonance (LF-NMR)

A MesoMR nuclear magnetic resonance food analysis and imaging system (Shanghai Newmax Technology Co., Ltd., Shanghai, China) was used for the measurement. A certain weight of hairtail surimi gel was wrapped in plastic wrap and put into a glass tube with a diameter of 25 mm. The T2 signal was acquired using nuclear magnetic resonance analysis software (Ver 1.0) and the CPMG sequence, and the sequence parameters were set as follows. Inherent parameters: RF signal frequency main value SF = 23 MHz, RF signal frequency offset O1 = 403,492.2 Hz, RF 90 degree pulse width P1 = 9 μs, RF 180 degree pulse width P2 = 17.52 μs; and Adjust parameters: When sampling, the receiver received a signal frequency range and signal sampling frequency SW = 250 KHz, repeat sampling interval TW = 4000 ms, RF delay RFD = 0.08 ms, analog gain RG1 = 20, digital gain DRG1 = 3, cumulative sampling times NS = 8, preamplification gain PRG = 1, number of echoes NECH = 12,000, and echo time TE = 0.12 ms. Each sample was repeated three times.

### 2.7. Total Sulfhydryl Content

A total sulfhydryl content detection kit (Nanjing Jianxing Bioengineering Institute, Nanjing, China) was used for detection. About 0.1 g of the sample was weighed and made accurate to 0.001 g, and we added 1 mL of extract to prepare a 10% homogenate, centrifuged it at 8000× *g* for 10 min at room temperature, and took the supernatant to be measured.

The 25 μmol/mL standard solution was diluted to 1, 0.5, 0.25, 0.00625, and 0.03125 μmol/mL standard solution with distilled water, and the sample was added as follows. We added 40 μL of the sample, 150 μL of reagent one, and 10 μL of H_2_O sample into the control tube. An amount of 40 μL of the sample, 150 μL of reagent one, and 10 μL of reagent two was added into the assay tube. The standard tube had 40 μL of standard substance, 150 μL of reagent one, and 10 μL of reagent two added. An amount of 150 μL of reagent one and 50 μL of H_2_O were added to the blank tube. We then mixed the mixture, placed it at room temperature for 10 min, and measured the absorbance value at 412 nm in a 96-well plate, denoted as Acontrol, Aassay, Astandard, and Ablank, respectively. The assay was performed as follows: ΔAstandard = Astandard − Ablank, ΔAassay = Aassay − Acontrol.

Drawing the standard curve: By taking the standard concentration of the standard solution as the x-axis and the ΔA standard as the y-axis, the standard curve was drawn to obtain the standard equation y = kx + b, and the ΔA measurement was substituted into the formula to obtain x (μmol/mL). The standard curve equation was y = 1.4829x + 0.0157, R2 = 0.9992
C = x/W(3)
where C is the total sulfhydryl content (μmol/g), W is the sample weight, and x is the concentration obtained by substituting it into the standard curve (μmol/mL).

### 2.8. Volatile Flavor

The volatile flavor compounds were detected using an HS-SPME-GC/MS analysis. The SPME extraction fiber head (50/30 μm PDMS/CAR/DVB (2 cm), Supelco Company, USA) was first aged at 250 °C at the GC-MS injection port until there were no stray peaks. About 10 g of the sample was transferred into a 15 mL sample bottle and sealed with a cap. The temperature of the solid phase microextraction device was set at 80 °C, and it was preheated for 15 min. The SPME extraction head was inserted into the headspace of the sample through the bottle cap, and the fiber head was pushed out. The headspace extraction was carried out for 50 min with the extraction head being 1.5 cm above the upper surface of the sample. We pulled back the fiber head and pulled out the extraction head from the sample bottle. Then, we placed the extraction head into the injection port of the GC-MS (7890A-5975C gas chromatograph, Agilent Technology Co., Ltd., Palo Alto, USA), pressed the fiber head out, and conducted the analysis at 250 °C for 5 min. The chromatographic conditions were HP-INNOWAX (60.0 m × 250 μm, 0.25 μm). The initial temperature of the column was 40 °C (maintained for 5 min), and the temperature was increased to 230 °C (maintained for 10 min) at a rate of 5 °C/min. Gasification chamber temperature 250 °C; transmission line temperature 230 °C; carrier gas He; carrier gas flow rate 1.0 mL/min; No split injection was performed. Ms conditions: EI source; electron energy 70 eV; ion source temperature 230 °C. Quadrupole 150 °C; the scan mode was “Scan”. The scanning quality ranged from 35 to 500 amu. Qualitative analysis: performed to detect whether the ingredient is qualitative with the MS database NIST11 and retention time. Quantitative analysis: The area normalization method was used for the quantification.

### 2.9. Electronic Nose

The methods used by Tian [38] were referred to and slightly modified. A PEN3 electronic nose (AIRSENSE Analytics GmbH., Schwerin, Germany) was used for measurement. The 10 sensors are W1C (aromatic compounds), W5S (nitrogen oxides), W3C (amines, aromatic compounds), W6S (hydrocarbons), W5C (alkanes, aromatic compounds), W1S (methane (methyl group)), W1W (inorganic sulfide), W2S (broad range alcohols), W2W (organic sulfide), and W3S (long chain) Alkanes). A sample of 5 g of minced surimi gel was placed in a 100 mL glass bottle and incubated for 30 min at room temperature (25 °C) to achieve headspace equilibrium. The flow rate of the sensing chamber was 300 mL/min, the flow rate of the measured sample was 300 mL/min, the test time was 60 s, the headspace temperature was 25 °C, and the sensor system was purged with clean air after each sample for 100 s until the signal returned to the baseline level. Each sample was tested three times in parallel, and the time point at which the sensor was at the most stable signal was taken for analysis. The data were analyzed using a principal component analysis (PCA), sensor loadings analysis (loadings analysis) (LA) and linear discriminant analysis (LDA) using the built-in software (WinMuster, ver 1.6.2.16).

### 2.10. Sodium Dodecyl Sulfate-Polyacrylamide Electrophoresis (SDS-PAGE)

A reference to and slight modification of the Liu et al. [32] approach was conducted. The MFHS was mixed with three times the amount of solution A (20 mmol/L Tris-HCl buffer (pH 7.5) containing 0.1 mol/L of NaCl and 0.02% NaN_3_), homogenized, washed at 4 °C for 15 min, and centrifuged at 5000 r/min for five minutes. The precipitate was combined with 2.5 times the volume of solution B (containing 0.45 mol/L of NaCl, 5 mmol/L of β-mercaptoethanol, 0.2 mol/L of magnesium acetate (Mg(CH_2_COO)_2_) and 1 mmol/L of EGTA to produce 20 mmol/L of Tris-HCl buffer (pH 6.8). After stirring for 1 h at 4 °C, the supernatant was diluted by adding twice the volume of 1 mmol/L of NaHCO_3_ solution, and we placed the mixture at 4 °C for 15 min and centrifuged it at 9500 r/min for 10 min. We poured out the supernatant and dissolved the precipitate in 2 mL of solution C, which contained 0.5 mol/L of NaCl and 5 mmol/L of Tris-HCl buffer (pH 7.5); the concentration of myofibrillar protein in the solution was determined using a biuret assay.

We tuned in the solution using a solution C protein at a concentration of 1.5 mg/mL; we placed 15 μL on the specimens from a buffer (5×), combined them in 60 μL of protein solution, and blended them for 5 min at a boil at 10,000 r/min. After cooling to room temperature, we centrifuged for 10 min and took 10 μL of the supernatant for electrophoresis. Conditions for electrophoresis: 4% concentrate glue; 7.5% separate glue; and voltage for the concentrate glue section set at 80 V, then adjusted to 120 V until the conclusion of the separate glue operations. The samples were dyed overnight with colloidal dye (blue silver technique) before being decolored with distilled water. The samples were photographed and examined using a gel imaging device after three water changes.

### 2.11. Raman Spectrum

The Nannan Yu et al. technique for performing confocal Raman spectroscopy was used [39]. Tin foil was used to secure the sample to a glass slide. A microscope (HORIBA ABX SAS., Parc Euromédecine, France) with a 50× lens was used to focus an excitation laser beam (633 nm excitation line of a Spectra Physics Ar laser) on the sample, and the Raman signal was recorded in the backscattering direction. Samples were taken between 400 and 3500 cm^−1^. The following parameters were used for each spectrum: 1 scan, 30 s exposure duration, 2 cm^−1^ resolution, 2 min each spectrum, 120 cm^−1^/min sampling rate, and data collection every 1 cm^−1^. Using Labspec version 5.0, the obtained spectra were baseline corrected and smoothed. All analyses were performed 3 times in parallel and averaged. The secondary structure of the protein is mainly located in the spectral frequency of the 1600 cm^−1^ to 1700 cm^−1^ region, and the spectral peaks at the wave numbers of 1600 cm^−1^ to 1640 cm^−1^, 1640 cm^−1^ to 1650 cm^−1^, 1650 cm^−1^ to 1660 cm^−1^, and 1660 cm^−1^ to 1700 cm^−1^ characterized the β-sheet, random coil, α-helix, and β-turn structure, respectively.

### 2.12. Analysis of Fungal Microbial Community Succession during Fermentation

A 50 mL chilled centrifuge tube containing 2 g of the material was weighed, 10 mL of sterilized phosphate buffer (pH 7.4) was added, and the tube was shaken at 4 °C for 2 h. The cell precipitates were then obtained after 15 min of spinning at 12,000× *g* and 4 °C. The resulting cell precipitates were combined by repeating this procedure 3 times. A DNA extraction tool was used to remove all of the bacteria’s DNA, and the samples of extracted DNA were then sent to Shanghai Personalbio Technology Co., Ltd. (Shanghai, China) for sequencing.

An appropriate amount of qualified genomic DNA was taken and interrupted by ultrasound according to the corresponding interrupted conditions and according to the size of the fragment. After centrifugation, the interrupted products were collected and purified. End Repair Reaction Buffer and EndRrep Enzyme Mix were added to the purified product, mixed and centrifuged, and placed into a PCR instrument for the end repair and addition of A; then, a connector and ligase were added to the connector ligation reaction. The ligation products were separated and purified by agarose gel electrophoresis followed by PCR amplification and purification to form a sequencing library. An illumina platform was used to sequence the qualified libraries.

Fungi sequencing primers: F:GGAAGTAAAAGTCGTAACAAGG R:GCTGCGTTCTTCATCGATGC; sequencing regions: ITS_V1; main methods: DADA2 (2016b); database: unite_8. The paired-end sequencing of community DNA fragments was performed on the Illumina platform.

Eliminate redundancy processing: Raw sequencing data were stored in FASTQ format (R1.fastq and R2.fastq, Read 1 and Read 2 sequences were paired one-by-one). The DADA2 method and de-primer, quality filtering, denoise, stitching, and de-chimerism techniques were applied.

Bioinformatics analysis and species annotation: The UNITE database (Release 8.0, https://unite.ut.ee/ (accessed on 4 February 2021)) was used for fungi. Adopting the QIIME2 classify–sklearn algorithm (2018b) (https://github.com/QIIME2/q2-feature-classifier (accessed on 4 February 2021)); For the characteristic sequences of each ASVs or representative sequences of each OTU, species annotation was performed using the pre-trained Naive Bayes classifier using default parameters in QIIME2(2019.4) software.

### 2.13. Statistics of Data

SPSS 27 was used for the statistical analysis of the experimental data, and multiple regression analysis was performed between the measured volatile substances of MFHS and the relative abundance of fungal microorganisms. A principal component analysis was performed using R software (RStudio 1.4), and a redundancy analysis (RDA) was performed using the Vegan package to analyze the effects of environmental factors on the microbial community structure. After normalizing the data, R software was used for heat mapping.

## 3. Results and Discussion

### 3.1. Physicochemical Properties

#### 3.1.1. Texture Profile Analysis (TPA)

The results of the determination of the texture properties of the hairtail surimi gel by Monascus fermentation at different times are shown in Figure 1. The fermentation time had significant effects on the texture of surimi gel (*p* < 0.05). The hardness, gumminess, and chewiness of surimi increased after fermentation for 1 h and then gradually decreased. Springiness did not change within 3 h of fermentation but significantly decreased after 3 h. The cohesiveness value gradually decreased, while the adhesiveness value significantly increased with the increase in fermentation time. Therefore, the 1 h fermentation group improved the texture properties of the hairtail surimi gel.

The hardness and elasticity indexes reflect the sensory palatability of surimi. The elasticity of surimi is generally related to the content of salt-soluble protein; in particular, it is directly related to the content of myosin. Myosin generally consists of myofibrillar proteins in the form of a thick filament structure of myofibrils. Figure 1 showed that the elasticity of hairtail surimi gel did not change within 3 h of fermentation time but decreased after 3 h. The gelation process of surimi myofibrillar protein is affected by many factors, such as metal ions, pH, enzymes, etc. [40]. Through the controlled experiments, it could be revealed that Monascus fermentation produced protease that was mainly the hydrolyzed myofibrillar protein, which unfolded the structure of myofibrillar protein to a degree and exposed the reaction groups in the molecule. It could promote the aggregation of protein–protein and the formation of a gel network, and it showed improvements in the hardness, adhesion, chewiness, and recovery of surimi gel, but has little impact on the myosin of the surimi myofibrils. However, with the extension of fermentation time, the proteases produced by fermentation further hydrolyze myofibrillar proteins after 3 h of fermentation, and myofibrillar proteins are composed of 55% to 60% myosin. This suggests that myosin may be decomposed by the influence of the enzyme produced by fermentation, which reduces the formation of the gel network structure and shows the lower quality of the surimi gel.

Adhesiveness is a term that refers to the force required to separate the tongue and teeth from the food surface when they are affixed (usually negative). Figure 1 demonstrates that Monascus fermentation significantly enhanced the adhesion of hairtail surimi gel, which considerably reduced the power of adhesion or the sticky taste of surimi when eating, improving the palatability of hairtail surimi. Myosin hydrolysis in surimi resulted in the structure of the gel network deteriorating as fermentation time increased, probably due to the continuous fermentation of Monascus. The hydrolysis-produced small molecular proteins clumped together through chemical pressures, which decreased the binding of the hairtail surimi gel.

#### 3.1.2. Results of the Determination of T-VBN in MFHS

The T-VBN of surimi at various Monascus fermentation periods is depicted in Figure 4a. During the 0–3 h of fermentation with Monascus, there was no discernible change in the T-VBN of hairtail surimi. The amount of hairtail surimi progressively rose during 3–5 h of fermentation, reaching a maximum of 25.4 mg/100 g at 5 h. Therefore, fermentation can inhibit the growth of spoilage microorganisms in surimi and reduce the formation of amines. This result is similar to the experimental results of ZHANG Da-wei et al. [41]. However, with the increase in time, the strains in this experiment became complex microorganisms. When the fermentation period approached 7 h, its T-VBN was 34.8 mg/100 g, which does not satisfy the criteria of the national standard (GB 10136–2015 T-VBN/(mg/100 g) ≤ 30). As a result, the time of the subsequent experiments of MFHS was limited to 5 h.

#### 3.1.3. Water Holding Capacity (WHC)

Studies have shown that fermentation can increase the water content of fish-like products, and the increase in the water holding capacity is mainly related to the retention of bound water and intermediate water [42,43,44]. The holding capacity of MFHS gel increased in 1 h and then gradually decreased (Figure 4b), which was mainly due to the promotion of Monascus fermentation for 1 h to form a compact, three-dimensional network structure; the holding capacity was 19.5% greater compared with the unfermented group. After 3 h, the holding capacity of the MFHS gel decreased.

#### 3.1.4. Microstructure Observation of Surimi Gel

The formation of the surimi gel network structure was closely related to the ordered aggregation. The more order and the closer the connection between the gel networks, the better the gel quality [45,46]. Monascus fermentation can promote the formation of surimi gel, allowing the gel network structure to be more compact and ordered and allowing the pore size distribution to be more uniform (Figure 2). The proteolytic enzymes produced by fermentation promote the unfolding of the myofibrillar protein structure, which exposes more protein binding sites and drives the cross-linking and interaction between proteins. This microporous structure presented by Monascus fermentation also enhanced the water holding capacity of the surimi gel. Compared with the unfermented group, the 1 h group displayed a more uniform and orderly gel network with small pore sizes and a more uniform distribution, while the 3 h group displayed large pores in the gel network. In addition, the 5 h group displayed local roughness of the gel network, which the structure gradually loosened; the pores grew larger, and the gel network was obviously fractured. The primary explanation is that the continuous Monascus fermentation causes the metabolically generated enzymes to further hydrolyze the proteins and their connecting bonds, causing the pores in the gel network structure to become larger and the structure to become looser. 

#### 3.1.5. Determination of Water Distribution in Surimi Gel

The T2 inversion spectrum of fermented hairtail surimi can be divided into three relaxation periods based on the distribution of peaks: segment a (0.5–4 ms), segment b (25–220 ms), and segment c (270–1011 ms) (Figure 3). The lateral relaxation time (T2) of protons varies, and they stem from the different chemical environment location. A shorter T2 will result in fewer degrees of freedom for the proton, and a longer T2 will result in more degrees of freedom for the proton. The size of T2 reflects the size of water freedom in the sample, where a longer T2 indicates that the water in the sample is less bound and is easier to be expelled [47,48,49].

Among the three peaks of T2, segment a has the shortest time; this part is bound water that most closely combined with other molecules and is considered to be the non-rotating “bound water” in the magnetic field. Segment b is semi-bound water, which is mainly composed of amide groups in protein molecules that combined with water at a small bond energy, and it is bound by the structure of the protein gel network; it is bound water that is next to free water and can easily transform, thus being considered as “bound” water that can rotate in the magnetic field. Segment c, which has the longest duration, consists of free water, which is mainly volume water and structural water; this part of the water has the molecular mobility of water in an aqueous solution.

The T2 time of hairtail in segment a shifted to the left compared with the unfermented group, and the peak value and peak area did not change remarkably. This indicates that after the Monascus fermentation, the water molecules more closely combined with the polar groups on the surface of the protein. The possible factor that could explain this is that the enzymes and metabolites produced by fermentation hydrolyzed the myofibrillar protein and opened the helix, with the closer surimi binding with water occurring through the combination of hydrogen chains; this is also consistent with the results of the partial fermentation stage of mandarin fish [44]. The peak value and peak area of segment b were significantly larger. The T2 time shifted to the left, and the T2 value decreased compared with the unfermented group. This indicates that Monascus fermentation increased the peak and peak area of non-flowing water, which promotes the tight formation of a three-dimensional network gel structure of hairtail surimi with a more uniform pore size, which consists of bound water in the gel network and reduced water loss. By comparing the peak value and peak area of segment b of surimi at four fermentation times, it can be seen that with the increase in fermentation time, the peak value and peak area of hairtail surimi first increased and then decreased. The 1 h fermentation group was the highest, which revealed the 1 h fermentation group as having the highest semi-bound water content and best WHC. At the same time, it can be seen from the Figure 3c segment that the peak value and peak area of segment c in the Monascus fermentation group decreased and had the peak shift right, indicating that the free water content of the Monascus fermentation group was lower than the unfermented group but also that the binding was stronger than the unfermented group; the results reflect that fermentation reduces the water activity of surimi, which is consistent with the decrease in water activity of fermented mandarin fish [43]. The peak value and peak area of segment c were the smallest in the 1 h fermentation group. As the fermentation time went on, the free water content of segment c first decreased and then increased. There was no significant difference in the free water content between the 5 h fermentation group and the unfermented group, but it was slightly longer at T2. In conclusion, Monascus fermentation improved the binding degree of hairtail surimi bound water and free water and enhanced the WHC of hairtail surimi gel.

#### 3.1.6. Determination of Total Sulfhydryl Content in MFHS

The total sulfhydryl content of surimi significantly decreased after Monascus fermentation (*p* < 0.05) (Figure 4c). During the gelation of surimi, myofibrillar proteins unfolded to expose free sulfhydryl groups, which were oxidized to disulfide bonds to promote the formation of a gelation network. According to the above analysis results, Monascus fermentation can promote the formation of the gel network structure of surimi. Additionally, the formation of disulfide bonds decreases the free sulfhydryl content and increases the gel network structure; however, the total sulfhydryl content does not significantly change. Furthermore, the total sulfhydryl content in the surimi decreased, which may be related to the protease produced during the Monascus fermentation. A significant analysis showed that there was no distinct difference in the content of total sulfhydryl groups between the unfermented group and the 1 h group. The main reason was that the protease produced by Monascus fermentation in the 1 h group promoted the unfolding of the myofibrillar protein structure, and the exposed sulfhydryl groups were oxidized to disulfide bonds to form the gel network structure of hairtail surimi. Furthermore, the hairtail surimi network structure was hydrolyzed and the disulfide bond was destroyed under the continuous action of proteases, which resulted in a significant reduction in the total sulfhydryl groups in surimi after 3 h and 5 h of fermentation. This phenomenon was also consistent with the gel quality decrease after 3 h of fermentation; it is also one of the reasons for the better gel quality of the surimi in the 1 h Monascus fermentation group that we mentioned above.

#### 3.1.7. Determination Result of SDS-PAGE

Myosin is the most important myofibrillar protein in the formation of the surimi gel network, and it contains six polypeptide units, two myosin heavy chains (MHCs), and four myosin light chains (MLCs) that are responsible for the formation of protein gel networks. MHC (220 kDa), tropomyosin (TM, 38 kDa), troponin T (TNT, 35 kDa) and MLC (25 kDa) are the primary bands of surimi (Figure 5). After Monascus fermentation, the surimi protein molecules were degraded, and the TM and MLC and MHC bands were altered. MHC is more readily degraded by proteolysis, which is commonly caused by endogenous proteases that are in hairtail surimi and proteases that are produced by microbial fermentation [50,51,52]. Figure 5 illustrates how fermentation encourages the growth of myofibrillar protein molecules. At 1 h, the pigment of the MHC and TM band is darker than that of the other groups; at 5 h, as tropomyosin and myosin levels rise, the MHC band becomes lighter. Meanwhile, the 25–35 kDa small molecule protein or peptide increases. Figure 5 showed that Monascus fermentation encourages the expansion of the myofibrillar protein structure during 1 h of fermentation and produces more small molecule proteins to promote the formation of the surimi gel network. The MHC strip became shallow as time continued, and its degradation prevented surimi from being fully crosslinked during the gel process. Correspondingly, this makes the three-dimensional network structure that is formed change from dense to loose and porous, which reduces the quality of surimi gel.

#### 3.1.8. Determination of Myosin Secondary Structure in Surimi

With the increase in fermentation time, the content of the protein α-helical structure with the ribbon was changed differently; the content of the β-fold gradually decreased, and the content of the β-corner first increased and then decreased, but an irregular curl showed the opposite change (Figure 4d). The myosin molecular chains are unfolding, which make the myosin transform from an α-helical structure to a β-fold, β-corner, and irregular curl structure. Compared with the unfermented group, the content of the α-helical structure of myosin in the fermented 1 h group decreased to 0, the content of the β-corner structure increased, and the content of the β-folded and irregular curl structures decreased. The main reason is related to protease hydrolysis causing a 180° inflection of the peptide bond skeletons, thus changing the orientation and spatial conformation of the peptide chain. In addition, the formation of the β-turn and random coil is beneficial to the subsequent cross-linking of proteins in surimi at high temperature and enhancement of the gel network structure of surimi [53,54]. At 3 h, the protein structure of surimi was mainly an α-helix and β-turn. It might be that the extension of fermentation time promoted the gelation process of surimi proteins and formed a loose gel network structure. When at 5 h, the secondary structure of surimi protein was mainly a β-turn, β-fold, and random coil structure. A possible reason is that the network structure of surimi protein was hydrolyzed by the protease that was generated by fermentation, which further destroyed the α-helix structure and changed the structure to comprise a greater number of random coils and a greater portion of the fold. The growth of the random helix structure showed that the protein’s previously orderly structures gave way to loose, disordered ones. The surimi protein’s degree of denaturation was increased, and the gel network structure that was created was poor. 

### 3.2. Determination of Volatile Substances

#### 3.2.1. Determination of Volatile Substances in Surimi Gel by Electronic Nose

The total contribution rate of the first principal component (PC1) and second principal component (PC2) of the volatile components of hairtail surimi gel at different fermentation times was 99.58% (>85%), which indicates that the two principal components fully covered the main information characteristics of the sample and that the distribution characteristics of each group’s data on PC1 were the main factor in determining the discrimination effect. The surimi samples with different Monascus fermentation times showed a certain change trend in PC1; meanwhile, the four groups could be separated, which indicates that the volatile flavor was markedly distinct with different Monascus fermentation times. Moreover, the unfermented group was similar to the 1 h fermented group but different from the 3 h and 5 h groups, which indicates that the characteristic volatile flavor substances of the Monascus-fermented hairtail surimi significantly changed from 1 h of fermentation (Figure 6a).

The loading analysis results of the four groups of samples are shown in Figure 6b, which shows that the contribution rate of PC1 is 98.69%; the larger that the main response value of each sensor on PC1 is, the more effective it is for sample identification. Additionally, the sensor was distributed closer to the origin (0,0) and has a smaller ability to identify the sample. The results showed that the response value of W1W (inorganic sulfide) on PC1 was the largest, and it is the main sensor for identifying the MFHS gel. The response values of W1S (methane (methyl)) and W5S (nitrogen oxide) on PC2 were relatively large, and the contribution rates of other sensors (W3S, W6S, W5C) to the first and second principal components were almost zero, indicating that the three sensors could hardly identify the volatile flavor of hairtail surimi gel at different fermentation times. To clarify the changes in the volatile flavor components of hairtail surimi gel during Monascus fermentation, GC-MS was used to further analyze and identify the volatile flavor components of hairtail surimi gel.

#### 3.2.2. Results of GC-MS Measured Volatile Substances in Hairtail Surimi

A total of 138 volatile substances were detected using GC-MS technology, including 20 alcohols, 7 aldehydes, 12 olefins, 4 phenols, 12 alkanes, 8 ketones, 15 esters, 6 acids, 16 benzenes, 4 ethers, and 8 amines, as well as 26 other types of compounds (Table 1). The flavor of the MFHS gel is significantly impacted by these variations in volatile chemicals. Statistics on the quantity and relative content of different volatile substances at distinct fermentation times revealed that Monascus fermentation increases the types of benzene and other volatile substances while decreasing the types of aldehydes, olefins, phenols, alkanes, and ethers in the surimi gel’s volatile substances. Moreover, the kinds of alcohols and amines show a trend of first decreasing and then increasing with the fermentation period, and acids and esters exhibit an opposite tendency. The volatile substances of MFHS were mainly alcohols, esters, benzenes, aldehydes, and other classes. Moreover, the relative content of each category of volatile flavor exhibited various variations with the change in fermentation time. Compared with the changes in the volatile substances of hairtail surimi gel with Monascus fermentation at different times, we further examined the volatile substances that were reduced, increased, and added after fermentation. 

After Monascus fermentation, there were 26 kinds of volatile substances in the surimi gel that were significantly reduced, including octanol, trans-2-octen-1-ol, valeraldehyde, heptaldehyde, etc., and 17 kinds of volatile substances in the fermentation process that first decreased and then increased.

Aldehydes may be generated by lipid oxidation, which is connected to the grassy and fatty scent of aquatic items, and aldehydes have a strong odor superposition effect [55,56,57]. Heptaldehyde is an aldehyde that has an oil oxidation odor that is similar to a fishy smell [58]. Heptaldehyde was destroyed in the surimi gel during Monascus fermentation. Nonanal has a fishy smell, and its content was eliminated within 3 h of fermentation; at 5 h, the relative content was 42.8% of the control group. Hexanal, which has a fishy odor, a grassy flavor, and other smells, may be generated when n-6 fatty acids are oxidized. According to the research, it is thought to be the primary culprit behind fish’s beany flavor [7]. The high content of hexanal is often mixed with volatile compounds of C8 and C9 to contribute to the aroma of fish. The results showed that the relative content of hexanal was reduced, and it was reduced to its lowest at 1 h of fermentation, where it represented 41.0% of the blank group. Therefore, Monascus fermentation effectively reduces the fishy flavor of surimi through the elimination or reduction of fishiness-related aldehydes.

1-octene-3-ol in alcohols has been reported as a fishy component of fish [59]. It can be seen from the Table that when the fermentation is 0–3 h, the relative content of 1-octene-3-ol drops to 0; at 5 h, it represents 75.9% of the blank group. Octanol has an earthy, metallic taste, and trans-2-octene-1-ol is a potential key substance for fishy odor [60,61]. These two alcohols were eliminated during the Monascus fermentation. Alcohols are often believed to be produced from fatty acids or reduced from carbonyl compounds, and 1-octen-3-ol is a hydroperoxide of linoleic acid breakdown, which results in a mushroom-like odor that is common in volatile fish components. It is believed that 3-methyl-1-butanol concentrations rise as a result of microbial deterioration during fish preservation [62]. Oleic acid was oxidized to produce octanol. Furthermore, alcohols that diminish hairtail surimi after fermentation include cis-2-penten-1-ol and phenylethanol, which are also present in related fish volatile taste compounds.

Among the esters, 2-ethylfuran is regarded as a fishy flavor element that is primarily present in the volatile flavor of freshwater fish. It is produced by amino acids and sugars via the Maillard and Strecker degradation reaction and helps give some foods flavors such as burnt, sweet, bitter, cooked meat, and coconut [62]. The relative concentration of 2-ethylfuran was 44.3% and 50.4% of the control group at 1 h and 3 h, respectively, which decreased the fishy smell of the hairtail surimi gel. In general, the fishy substances in the volatile flavor of hairtail surimi gel that was fermented for 1 h and 3 h after Monascus fermentation were significantly eliminated or reduced, which was conducive to improving the flavor of the hairtail surimi.

After Monascus fermentation, the volatile flavor of the hairtail surimi gel was enhanced by the presence of 15 new volatile substances, 5 different types of volatile substances (ethanol, 1-penten-3-ol, n-amyl alcohol, benzaldehyde, and naphthalene), and 9 additional volatile substances in the late fermentation stage. Meanwhile, Monascus fermentation increased the variety and content of amines in the hairtail surimi to a certain extent. The newly added volatile substances were mainly methylbenzene (eight kinds), and the inclusion of methylbenzene improved the taste of the hairtail surimi gel.

The content of aromatic aldehydes benzaldehyde was 312.5% of the blank group during fermentation for 3 h, which has an almond flavor. Isovaleraldehyde is thought to be the volatile main flavor of cod, pomfret, and other fish. The threshold of the substance is only 0.4 μg/kg, and it is comparable to apple fragrance, sweet rolls, chocolate, and cocoa-like aromas at a high concentration. Ans present a peach-like flavor when the concentration is less than 10 mg/kg; this ingredient enhances the flavor of hairtail surimi gel.

Ethanol, 1-penten-3-ol, n-amyl alcohol, and 2-octanol are some of the alcohols that are produced after Monascus fermentation. Ethanol is a metabolite during Monascus fermentation that increases with the increase in fermentation time. Meanwhile, 1-penten-3-ol is a metabolite of microorganisms with mushroom flavor that is a typical flavor compound in sardines, white silica fish, and silk fish; some also believe it to be the main component of fishy smell [63]. The typical fish taste of the hairtail surimi gel was somewhat enhanced by the longer fermentation period; the fermentation at 0–5 h was 90.9%, 123.2%, 148.2%, and 187.7% of the blank, respectively. N-amyl alcohol has an oily, sweet, and clear flavor. The relative content of 1 h–5 h was 131.1%, 127.0%, and 179.7% of the blank group, respectively, which improved the flavor of the surimi gel. 2-octanol is a key aromatic component in many foods with a fruity and fresh aroma. It is produced after fermentation for 5 h and enriches the flavor of surimi. In general, Monascus fermentation enhanced the almond and sweet taste, increased the aromatic flavors such as fruit and fragrance, and enriched the flavor of the hairtail surimi.

### 3.3. Microbial Diversity

The fungal flora in MFHS are mainly *Ascomycota*, *Basidiomycota*, *Mortierellomycota*, *Mucoromycota*, *Chytridiomycota*, *Basidiobolomycota*, *Neocallimastigomycota*, and *Glomeromycota*; *Ascomycota* and *Basidiomycota* are the dominant microorganisms. *Ascomycota* increased from 10.16% to 90.78%, and Basidiomycota decreased from 76.15% to 7.55% after the Monascus addition. The addition of Monascus changed the dominant fungal microorganisms in the raw surimi, and their relative abundance fluctuated with the fermentation time (Figure 7a).

At the genus taxonomic level, according to the species annotation and abundance information of all samples at the genus level, the top 30 fungal species in the surimi were selected for analysis (Figure 7b). The fungal flora in the blank group were mainly *Haglerozyma* (59.36%), *Cryptococcus* (4.70%), *Cutaneotrichosporon* (4.39%), *Apiotrichum* (2.97%), *Candida* (1.98%), and *Trichosporon* (1.18%). After adding *Monascus*, the fungal flora in surimi were mainly *Monascus* (89.95%) and *Haglerozyma* (6.60%), and the relative abundance of *Haglerozyma* gradually decreased to 0.28% after 5 h. At the same time, the original genera *Cryptococcus*, *Cutaneotrichosporon*, *Apiotrichum*, *Candida*, and *Trichosporon* were inhibited and remained extremely abundant during fermentation. It can be concluded that the genus *Monascus* dominates the activity of fungal microorganisms during the Monascus fermentation, and the addition of *Monascus* inhibits the activities of *Cryptococcus*, *Cutaneotrichosporon*, *Apiotrichum*, Candida, and *Trichosporon*.

### 3.4. Correlation Analysis between Fungal Microbiota and Texture

To further research the relationship between the dominant microbial community and the volatile flavor, the relative content of the dominant bacterial community at the genus level and the 10 types of volatile flavor were analyzed by a Spearman correlation, and a heat map was drawn (Figure 8a). The figure shows that there are 10 types of dominant fungal flora that are associated with 10 major classes of volatile flavor substances (*p* < 0.05). Among these, the microorganisms that significantly correlated with esters are the most, including *Monascus*, *Candida*, *Echria*, *Botryotrichum*, and others. The microorganisms that are associated with benzene are *Malassezia*, the associated with alcohol-flavored substances are *Trichosporon*, the aldehyde-related microorganisms are *Apiotrichum*, the olefin-related microorganisms are *Aspergillus*, the acid-related microorganisms are *Trichosporon*, and the amine-related microorganisms are *Fusarium*.

We created the heat maps by using the results of the Spearman correlation analysis on the volatile taste components with the reduced flavor of surimi following fermentation with fungal flora (Figure 8b). A total of 16 fungal microorganisms were significantly associated with the reduced volatile flavor of surimi, among which *Monascus* and *Echria* were significantly related to the volatile flavor substances of p-xylene, methoxy-phenyl-oxime, and phenyl. P-xylene was reported as the “fresh meat taste” and fishy smell of aquatic raw fish [64]. In addition, the microorganisms that were significantly associated with hexanal reduction were *Trichosporon* and *Apiotrichum*; those associated with paraxylene reduction were *Monascus* and *Echria*, that associated with the reduction of 1-pentene-3 alcohols was *Aspergillus*; and those associated with 1-octene-3 alcohol reduction were *Apiotrichum* and *Malassezia*. Hexanal has a fishy smell, grassy smell and other flavors, which may be produced by n-6 fatty acid oxidation. Studies suggest that hexanal is the main substance that causes fishy smell, and its content was high. It is often mixed with volatile substances of C8 and C9 to contribute to the flavor of fish [65].

Heat maps were created using the results of the Spearman correlational study on the volatile taste components with the increased flavor of surimi following fermentation with fungal flora (Figure 8c). *Monascus*, *Haglerozyma*, *Trichosporon*, *Cryptococcus*, *Cutaneotrichosporon*, *Apiotrichum*, *Candida*, and *Malassezia* spp. were the eight fungal microorganisms that were significantly related to the volatile flavor substances added after fermentation, and all of them were significantly associated with nonanal. Nonanal originates from the oxidation of unsaturated fatty acids, and it is the product of the microbial enzyme [66]. Trichosporon is highly related to amine volatile substances, indicating that it is mainly responsible for the production of amines in surimi. Trichosporon can synthesize lipase, protease, and other enzymes. Amino acids produced by hydrolyzing proteins, especially arginine, can be converted into ornithine or agmatine, which are indirect precursors of putrescine [66].

## 4. Conclusions

Hairtail surimi has a limited range of applications because of its weak gel quality and strong fishy flavor. It is a challenge for the seafood industry to raise the overall quality of hairtail surimi. This study aimed to investigate the effects of Monascus fermentation on the quality, microbial diversity, and volatile components of hairtail surimi. The gel quality and WHC of surimi may be strengthened by Monascus fermentation for 1 h, and the Spearman correlation analysis reveals that 16 fungal microorganisms were substantially connected with the diminished volatile taste of MFHS. *Monascus* and *Echria* were substantially correlated with the volatile taste compounds of p-xylene, methoxy-phenyl-oxime, and phenol, whereas *Trichosporon* and *Apiotrichum* had a significant link with hexanal reduction. The microorganisms that were significantly associated with paraxylene reduction were *Monascus* and *Echria*; that associated with the reduction of 1-penten-3 alcohols was Aspergillus; and those associated with 1-octen-3 alcohol reduction were Apiotrichum and Malassezia. There were eight fungal microorganisms that were significantly related to the volatile flavor substances added after fermentation, which were significantly related to nonanaldehyde, namely *Monascus*, *Haglerozyma*, *Trichopsporon*, *Cryptococcus*, *Cutaneotrichosporon*, *Apiotrichum*, *Candida*, and *Malassezia* spp.; *Trichopsporon* is highly related to amine volatile substances, which indicates that it mainly causes the production of amines in surimi. In summary, Monascus fermentation can raise and improve the aromatic taste, decrease or remove elements that provide a surimi fishy smell, and enhance the flavor of the surimi gel. Moreover, further studies should focus on the quality changes of hairtail surimi that is fermented with certain fungi in *Monascus* to develop the applicable strains and establish standard industrial production techniques for hairtail surimi.

## Figures and Tables

**Figure 1 foods-12-02891-f001:**
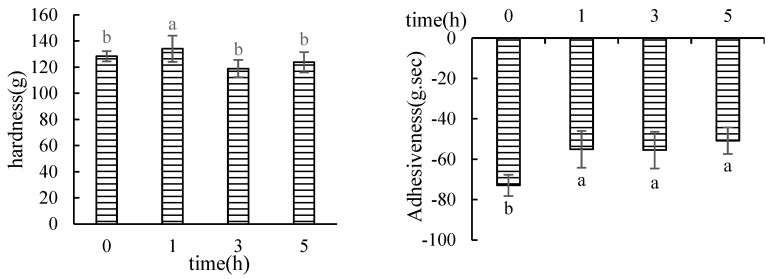
Texture determination of MFHS gel at different fermentation time (Different lowercase letters represent significant differences (*p* < 0.05)).

**Figure 2 foods-12-02891-f002:**
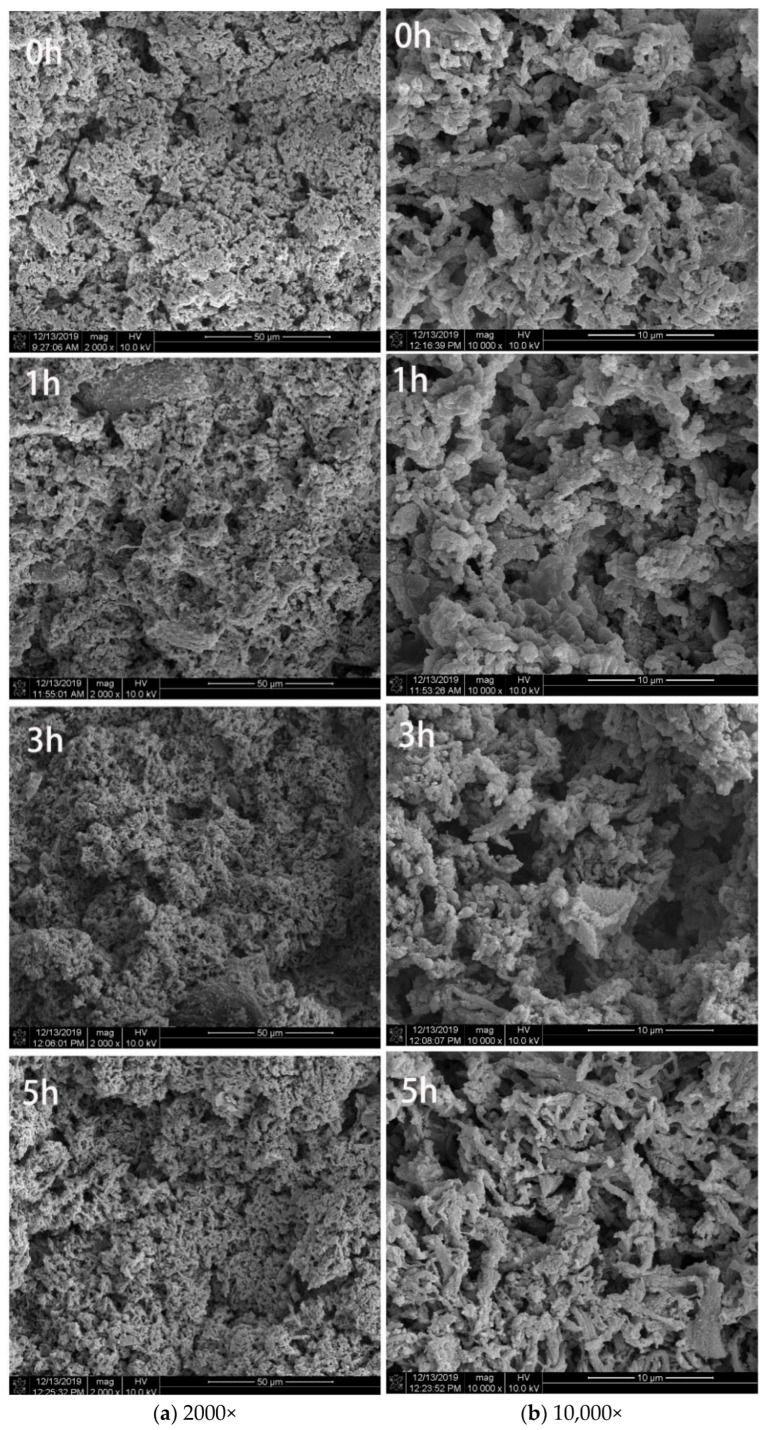
Scanning electron microscopic images of MFHS gel at different fermentation times.

**Figure 3 foods-12-02891-f003:**
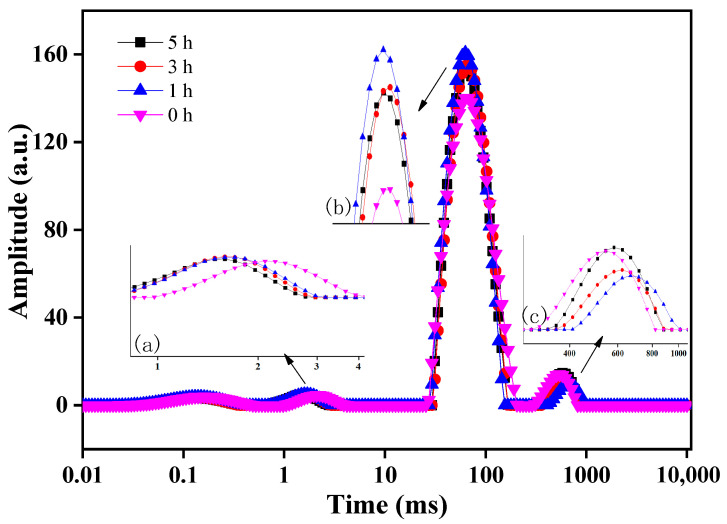
Inversion spectrum of the transverse relaxation time (T2) of MFHS at different fermentation times. (**a**–**c**) represent different relaxation time periods.

**Figure 4 foods-12-02891-f004:**
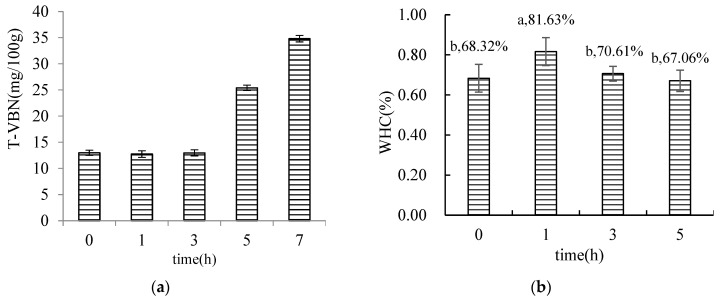
Effect of Monascus fermentation on TVB-N, WHC, total sulfhydryl content, and myosin secondary structure contention of hairtail surimi (fermentation temperature 25 °C; different lowercase letters represent significant differences (*p* < 0.05)).

**Figure 5 foods-12-02891-f005:**
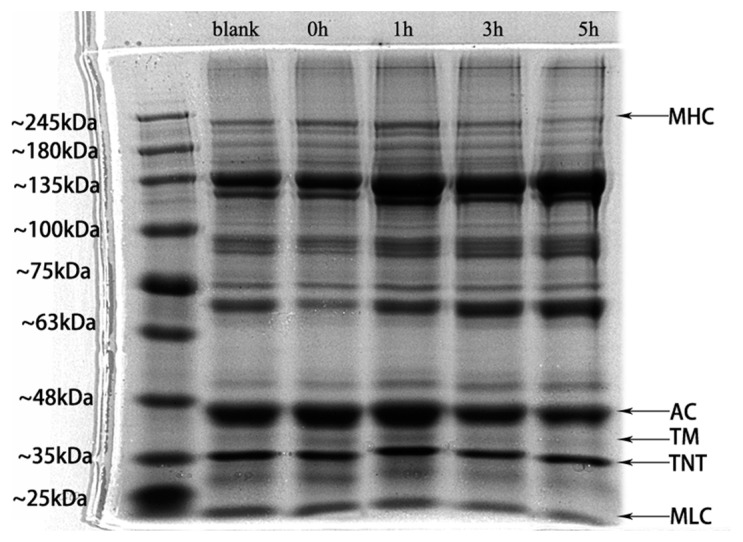
SDS-PAGE profile of protein in MFHS gel.

**Figure 6 foods-12-02891-f006:**
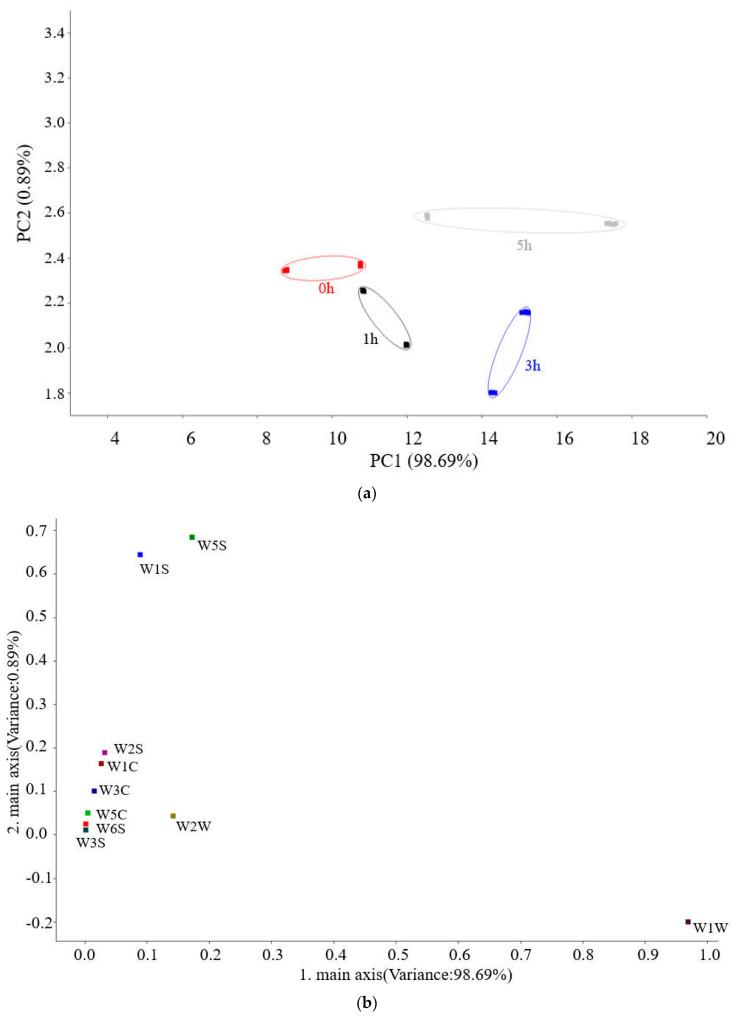
PCA, loadings analysis of the volatile flavor of hairtail surimi gel at different fermentation times ((**a**)—PCA analysis; (**b**)—Loading analysis).

**Figure 7 foods-12-02891-f007:**
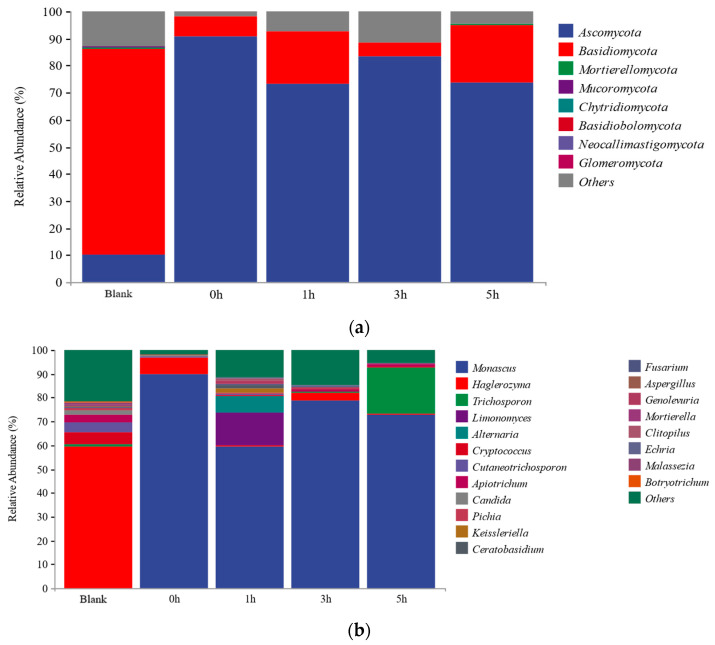
Relative abundance levels of fungal community in MFHS. ((**a**)—phylum, (**b**)—genus).

**Figure 8 foods-12-02891-f008:**
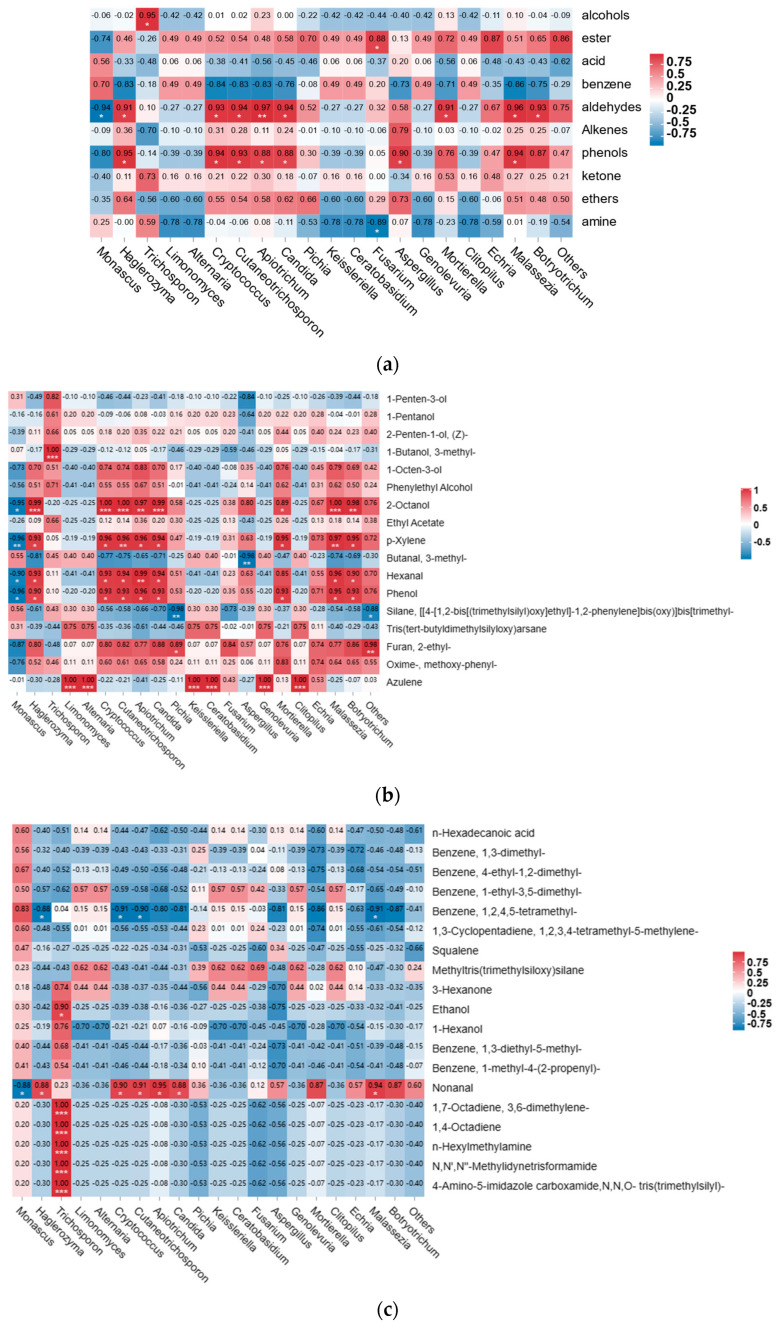
Correlation between the relative abundance of fungal genera and the volatile flavor of the MFHS gel. ((**a**) Correlation between the types of volatile flavor substances and the fungal genera; (**b**) correlation between the reduced volatile flavor substances and the fungal genera; (**c**) correlation between the increased volatile flavor substances and the fungal genera. * = significant at *p* < 0.05. ** = significant at *p* < 0.01. *** = significant at *p* < 0.001).

**Table 1 foods-12-02891-t001:** The relative contents of volatile compounds in hairtail surimi during fermentation.

Number	Compounds	CAS	Relative Content/%				
			Blank	0 h	1 h	3 h	5 h
1	Ethanol	000064-17-5	2.64	2.04	3.48	6.57	12.06
2	1,3-Butanediol, (S)-	024621-61-2	nd	0.39	nd	nd	nd
3	1-Penten-3-ol	000616-25-1	2.2	2	2.71	3.26	4.13
4	Piperidine-4-ol, 1,2,5-trimethyl-4-(2,4,5-trimethylphenyl)-	1000259-69-6	nd	0.59	nd	nd	nd
5	1-Hexanol	000111-27-3	1.52	1.38	nd	3	4.2
6	1H-Inden-5-ol, 2,3-dihydro-	001470-94-6	nd	0.59	nd	nd	nd
7	Ethanol, 2-(vinyloxy)-	000764-48-7	nd	0.76	nd	nd	nd
8	2-[2-[2-[2-[2-(tert-Butyldimethylsilyloxy)ethoxy]ethoxy]ethoxy]ethoxy]ethanol	1000352-10-3	nd	0.39	nd	nd	0.68
9	1-Pentanol	000071-41-0	0.74	nd	0.97	0.94	1.33
10	2-Penten-1-ol, (Z)-	001576-95-0	0.64	nd	0.55	0.53	0.89
11	Ethanol, 2-[2-(2-butoxyethoxy)ethoxy]-	000143-22-6	nd	nd	0.54	nd	nd
12	(3-Methyl-oxiran-2-yl)-methanol	1000194-22-9	nd	nd	nd	0.75	nd
13	Silanol, trimethyl-	001066-40-6	nd	nd	nd	nd	1.07
14	1-Butanol, 3-methyl-	000123-51-3	1.18	nd	nd	nd	8.79
15	2-Octanol	000123-96-6	nd	nd	nd	nd	1.01
16	1-Octen-3-ol	003391-86-4	3.81	nd	nd	nd	2.89
17	Phenylethyl Alcohol	000060-12-8	1.49	nd	nd	nd	1.67
18	2-Octanol	000123-96-6	2.27	nd	nd	nd	nd
19	2-Octen-1-ol, (E)-	018409-17-1	0.7	nd	nd	nd	nd
20	Cyclobutaneethanol, .beta.-methylene-	116203-80-6	1.18	nd	nd	nd	nd
21	Ethyl Acetate	000141-78-6	2.33	0.63	1.63	2.53	3.15
22	2-Ethyl-1-hexanol, pentafluoropropionate	1000365-51-1	nd	1.44	nd	nd	nd
23	2-Ethyl-1-hexanol, trifluoroacetate	1000365-19-6	0.77	nd	nd	nd	3.05
24	Formic acid, hex-2-yl ester	1000368-21-9	nd	nd	1.07	nd	nd
25	Arsenous acid, tris(trimethylsilyl) ester	055429-29-3	nd	nd	2.06	nd	nd
26	Hexyl chloroformate	006092-54-2	nd	nd	2.01	nd	nd
27	2-Ethyl-1-hexanol, pentafluoropropionate	1000365-51-1	nd	nd	2.58	nd	nd
28	3-Hydroxymandelic acid, ethyl ester, di-TMS	1000071-88-9	nd	nd	1.27	nd	nd
29	Thiocyanic acid carbazol-3,6-diyl ester	040736-18-3	1.39	nd	0.98	0.98	nd
30	Silanol, trimethyl-, propanoate	016844-98-7	nd	nd	0.75	0.74	nd
31	Propanoic acid, pentyl ester	000624-54-4	nd	nd	nd	2.83	nd
32	1-Octyl trifluoroacetate	002561-21-9	nd	nd	nd	0.69	nd
33	Arsenous acid, tris(trimethylsilyl) ester	055429-29-3	5.95	nd	nd	1.2	nd
34	Dicyclopropylmethanol, chlorodifluoroacetate	1000376-25-1	0.84	nd	nd	nd	nd
35	Formic acid, heptyl ester	000112-23-2	1.41	nd	nd	nd	nd
36	.gamma.-Guanidinobutyric acid	000463-00-3	nd	0.54	nd	nd	nd
37	Pentanoic acid, 3-methyl-	000105-43-1	nd	2.84	nd	nd	nd
38	Octanoic acid	000124-07-2	nd	1.67	nd	nd	nd
39	n-Hexadecanoic acid	000057-10-3	nd	18.4	8.35	5.54	nd
40	19,19-Dimethyl-eicosa-8,11-dienoic acid	1000297-01-9	nd	nd	0.38	nd	nd
41	cis-5,8,11,14,17-Eicosapentaenoic acid	010417-94-4	0.54	nd	nd	nd	nd
42	o-Xylene	000095-47-6	nd	0.34	nd	nd	nd
43	p-Xylene	000106-42-3	14.14	nd	2.28	0.51	4.28
44	Benzene, 1,3-dimethyl-	000108-38-3	nd	1.49	nd	2.46	nd
45	Benzene, 1,1′-(oxydi-2,1-ethanediyl)bis[3-ethyl-	055044-09-2	nd	2.44	nd	nd	nd
46	Benzene, 4-ethyl-1,2-dimethyl-	000934-80-5	nd	4.51	1.24	2.74	nd
47	Benzene, 1-ethyl-2,3-dimethyl-	000933-98-2	nd	1.47	2.56	nd	5.44
48	Benzene, 1-ethyl-3,5-dimethyl-	000934-74-7	nd	1.21	2.03	1.89	nd
49	Benzene, 2-ethyl-1,3-dimethyl-	002870-04-4	nd	nd	2.59	nd	nd
50	Benzene, 2-ethyl-1,4-dimethyl-	001758-88-9	nd	nd	nd	nd	0.76
51	Benzene, 2,4-diethyl-1-methyl-	001758-85-6	nd	nd	1.27	nd	nd
52	Benzene, 1,2,4,5-tetramethyl-	000095-93-2	nd	9.63	11.48	17.22	10.62
53	Benzene, 1,2,3,5-tetramethyl-	000527-53-7	nd	3.86	nd	6.41	nd
54	Benzene, 1,2,3,4-tetramethyl-	000488-23-3	nd	nd	8.84	nd	nd
55	Toluene	000108-88-3	nd	nd	0.87	nd	nd
56	Benzene, 1,3-diethyl-5-methyl-	002050-24-0	nd	nd	nd	1.6	1.86
57	Benzene, 1-methyl-4-(2-propenyl)-	003333-13-9	nd	nd	nd	1.24	1.09
58	Butanal, 3-methyl-	000590-86-3	nd	0.53	2	1.87	2.14
59	Hexanal	000066-25-1	9.25	3.97	3.79	4.58	5.66
60	Benzaldehyde, 2,4-dimethyl-	015764-16-6	nd	0.69	nd	nd	nd
61	Benzaldehyde	000100-52-7	0.64	0.48	1.5	0.98	1.33
62	Nonanal	000124-19-6	1.66	nd	nd	nd	0.71
63	Pentanal	000110-62-3	1.05	nd	nd	nd	nd
64	Heptanal	000111-71-7	4.42	nd	nd	nd	nd
65	1,3-Cyclopentadiene, 1,2,3,4-tetramethyl-5-methylene-	076089-59-3	nd	3.09	2.11	5.17	nd
66	Squalene	000111-02-4	nd	12.09	nd	nd	nd
67	2-Pentene	000109-68-2	nd	nd	4.42	nd	nd
68	1,3,6-Octatriene, (E,E)-	022038-69-3	nd	nd	0.49	nd	nd
69	1-Methylcycloheptene	055308-20-8	nd	nd	nd	0.5	nd
70	1,7-Octadiene, 3,6-dimethylene-	003382-59-0	nd	nd	nd	nd	0.94
71	1,4-Octadiene	005675-25-2	nd	nd	nd	nd	0.57
72	2,4-Octadiene	013643-08-8	3.86	nd	nd	nd	nd
73	3,5-Octadiene, (Z,Z)-	007348-80-3	1.47	nd	nd	nd	nd
74	Cyclooctene, 3-ethenyl-	002213-60-7	0.85	nd	nd	nd	nd
75	1,4-Cyclooctadiene	001073-07-0	0.52	nd	nd	nd	nd
76	2-Hexene, 3,5,5-trimethyl-	026456-76-8	3.85	nd	nd	nd	nd
77	4-Aminophenol, N,O-bis(pentafluoropropionyl)-	1000364-80-0	nd	0.12	nd	nd	nd
78	3-Aminophenol, N,O-bis(pentafluoropropionyl)-	1000364-81-6	nd	0.81	nd	nd	nd
79	Phenol	000108-95-2	2.42	0.93	1.25	1.19	1.51
80	p-Cresol	000106-44-5	0.48	nd	nd	nd	nd
81	Silane, [[4-[1,2-bis[(trimethylsilyl)oxy]ethyl]-1,2-phenylene]bis(oxy)]bis[trimethyl-	056114-62-6	nd	0.48	0.41	nd	0.49
82	Heptane, 2,6-dimethyl-	001072-05-5	nd	0.75	nd	nd	nd
83	3,8-Dioxatricyclo[5.1.0.0(2,4)]octane, 4-ethenyl-	053966-43-1	nd	0.46	nd	nd	nd
84	Tris(tert-butyldimethylsilyloxy)arsane	1000366-57-5	nd	1.14	1.52	nd	nd
85	Methyltris(trimethylsiloxy)silane	017928-28-8	nd	nd	1.77	1.76	nd
86	Pentane, 1-chloro-	000543-59-9	nd	nd	nd	5.6	nd
87	Silane, chloroethylmethyl-	006374-21-6	nd	nd	nd	0.79	nd
88	6,6-Diethylhoctadecane	1000360-41-8	2.39	nd	nd	nd	nd
89	1-Methyl-2-methylenecyclohexane	002808-75-5	1.42	nd	nd	nd	nd
90	Heptadecane, 2,6-dimethyl-	054105-67-8	2.36	nd	nd	nd	nd
91	7-Oxabicyclo[4.1.0]heptane, 3-oxiranyl-	000106-87-6	1.52	nd	nd	nd	nd
92	Heptadecane	000629-78-7	0.86	nd	nd	nd	nd
93	3-Hexanone	000589-38-8	nd	nd	0.88	nd	1.22
94	2-Hexanone	000591-78-6	nd	nd	nd	0.84	nd
95	2H-1,4-Benzodiazepin-2-one, 7-chloro-1,3-dihydro-5-phenyl-1-(trimethylsilyl)-	055299-24-6	nd	nd	0.82	nd	nd
96	1(3H)-Isobenzofuranone, 6-(dimethylamino)-3,3-bis[4-(dimethylamino)phenyl]-	001552-42-7	nd	nd	0.24	nd	nd
97	2-Butanone	000078-93-3	nd	nd	nd	nd	1.4
98	2-Nonanone	000821-55-6	0.71	nd	nd	nd	0.51
99	3-Octanone, 2-methyl-	000923-28-4	0.48	nd	nd	nd	nd
100	2-Undecanone	000112-12-9	0.84	nd	nd	nd	nd
101	16-Methyl-heptadecane-1,2-diol, trimethylsilyl ether	1000336-70-6	nd	0.28	nd	nd	nd
102	Octaethylene glycol monododecyl ether	003055-98-9	nd	1.44	nd	nd	nd
103	2,5-Dihydroxyacetophenone, bis(trimethylsilyl) ether	1000352-83-7	1.22	nd	nd	2.1	nd
104	4-(2-Acetylamino-1-(trimethylsilyloxy)ethyl)phenol	1000373-43-5	1.26	nd	nd	nd	nd
105	Benzamide, N-[7-(1-hydroxypropyl)-2,3-dihydrobenzo[1,4]dioxin-6-yl]-4-methyl-	1000316-96-2	nd	0.39	nd	nd	nd
106	Hydrazinecarbothioamide, N-ethyl-	013431-34-0	nd	0.61	nd	nd	nd
107	Acetamide, 2-(2-hydroxyethoxy)-	000123-85-3	nd	0.72	nd	1	nd
108	Acetamide, N-(4-imidazo[1,2-a]pyrimidin-2-ylphenyl)-2-methoxy-	1000338-04-4	nd	0.69	nd	nd	nd
109	n-Hexylmethylamine	035161-70-7	nd	nd	nd	nd	1.16
110	N,N′,N″-Methylidynetrisformamide	004774-33-8	nd	nd	nd	nd	1.06
111	4-Amino-5-imidazole carboxamide,N,N,O- tris(trimethylsilyl)-	1000079-30-4	nd	nd	nd	nd	0.38
112	Benzenamine, 4-bromo-3-chloro-N-(4-methylthiobenzylydene)-	314283-74-4	1.4	nd	nd	nd	nd
113	o-Cymene	000527-84-4	nd	0.57	5.17	nd	nd
114	Furan, 2-ethyl-	003208-16-0	1.31	nd	0.58	0.66	
115	Furo[2′,3′:4,5]thiazolo[3,2-g]purine-8-methanol, 4-amino-6.alpha.,7,8,9a-tetrahydro-7-hydroxy-, [6aS-(6a.alpha.,7.alpha.,8.beta.,9a.alpha.)]-	016667-76-8	nd	nd	1.13	nd	nd
116	Benzofuran, 2,3-dihydro-2-methyl-	001746-11-8	0.54	nd	nd	nd	nd
117	5-[Cyano-(3,4-dimethyl-5-oxo-1,5-dihydro-pyrrol-2-ylidene)-methyl]-2,3,3-trimethyl-3,4-dihydro-2H-pyrrole-2-carbonitrile	1000186-14-3	nd	1.31	nd	nd	nd
118	7H-Dibenzo(a,g)carbazole	000207-84-1	nd	0.37	nd	nd	nd
119	1-(6-Methyl-benzothiazol-2-yl)-3-(4-methyl-benzoyl)-thiourea	131120-14-4	nd	1.17	nd	0.34	0.53
120	m-Tolualdehyde, thiosemicarbazone	005706-82-1	1.93	nd	nd	nd	nd
121	methoxy-phenyl-Oxime	1000222-86-6	3.42	2.13	2.95	2.49	3.25
122	Naphthalene	000091-20-3	1.73	1.53	3.19	3.4	3.01
123	Naphthalene, 2-methyl-	000091-57-6	nd	0.66	nd	nd	nd
124	1H-Indene, 1-methylene	002471-84-3	nd	1.85	nd	nd	nd
125	1H-Indene, 2,3-dihydro-4-methyl-	000824-22-6	nd	nd	0.99	nd	nd
126	Oxazole, 2,5-dihydro-5-(4-methylphenyl)-4-phenyl-	036879-73-9	nd	0.78	nd	nd	nd
127	2-Mercapto-4-phenylthiazole	002103-88-0	nd	nd	0.83	nd	1.02
128	13H-Dibenzo[a,i]carbazole	000239-64-5	nd	nd	nd	1.57	nd
129	.beta.-D-Xylo-Hexopyranosid-4-ulose, methyl 2,3,6-tri-O-methyl-, (2,4-dinitrophenyl)hydrazine	041545-25-9	nd	0.45	nd	nd	nd
130	Pyrazolo[1,5-a]pyrimidine, 2,7-dimethyl-5-phenyl-	1000267-29-6	nd	0.9	nd	nd	nd
131	Azulene	000275-51-4	nd	nd	0.86	nd	nd
132	Octanoic acid, silver(1+)	024927-67-1	nd	nd	0.61	nd	nd
133	3-Phenylindole	001504-16-1	nd	nd	nd	0.92	nd
134	2-Methyl-7-phenylindole	001140-08-5	nd	nd	nd	nd	1.5
135	Methyl d-glycero-.beta.-d-gulo-heptoside	1000130-15-2	nd	nd	nd	0.61	nd
136	2-Hydrazino-4,6-dimethylpyrimidine ditms peak 2	1000332-01-7	nd	nd	nd	nd	1.56
137	4H-3,1-Benzoxazine, 6,7-dimethoxy-2-(4-methoxyphenyl)-4-propyl-	1000327-27-7	nd	nd	nd	nd	1.09
138	N-(Trifluoroacetyl)-O,O′,O″-tris(trimethylsilyl)norepinephrine	1000072-26-3	0.39	nd	nd	nd	nd

## Data Availability

The data will be made available on request.

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
