# Peer review of "Physiochemical Quality, Microbial Diversity, and Volatile Components of Monascus-Fermented Hairtail Surimi"

_foods, 2023, doi:10.3390/foods12152891_

Round 1

Reviewer 1 Report

It is interesting paper about surimi fermentation, however it is not easy to read since need more structured. Beginning from the title that should be more concise and with more information about the target of the study. For example in Ln 16: it is said that the work study the “changes....” but it is not explain "where" until ln 22 that it is “during fermentation process”.

It is necessary place the work in a theoretical context, and enables the reader to understand and appreciate your objectives. Please, describe the importance of the study, providing a broad context but remove the ideas no important.

Please clarify along the manuscript between surimi, fermented-surimi and fermented-surimi gel (check ln 96, ln 108, ln 305-6, ln 341, 664, etc)

-Sometimes “rpm” and others “r/min”?

-Please check “))”, it should be “)” for example ln 217, 235, etc

-        “High-throughput sequencing technology” usually describe sequence DNA and RNA but not HS-SPME-GC/MS.

Author Response

(1) 

The study target was added in ln14.

“during fermentation” was added to explain the determination sample in Ln17.

(2) 

An overview of the results and significance of the study was added in ln26-30.

(3)

Clarified the “surimi, fermented-surimi,  and fermented-surimi gel”: “fermented-surimi” was uniformly represented by “MFHS”, and others were modified in 201 and the manuscript(marked red).

(4)

“rpm” has been changed to “r/min” in ln249.

Ln242 “((” has been changed to “)”, other “((” were right.

(5) 

“High-throughput sequencing technology” was used to determine the fungal microbial community in MFHS(2.11), and HS-SPME-GC/MS was used to determine the volatile flavor of MFHS gel(2.8).

Reviewer 2 Report

Review foods-2453762 

The work is meaningful and parameters from different aspects were well performed. However, the integration of the results from different parameters needs further improvement. In-depth discussion is needed to discuss more science behind these changes. 

The correlation discussion was superficial. This correlation should not only be data correlation but also the logic correlation among the parameters for a seafood system. The authors are suggested to search database like Web of Science with searching title words ‘microbiota composition’ AND ‘quality’ AND topic ‘seafood’, to get related reference(s) for further discussion. 

For surimi system, it is critical that the protein gelling properties were strongly affected by the calcium ion and the status. Therefore, to enhance the discussion, the authors are suggested to search database like Web of Science with searching title words ‘calcium ion’ AND ‘protein’ AND topic ‘surimi’, to get related reference(s) for further discussion. 

For seafood processing with time, the fermentation here in this report, for instance, the fermentation was not happening alone, myofibril protein as a key protein in surimi, experiencing degradation within the fish meat, fish fillet, for instance, which should be considered during the discussion. Therefore, to enhance the discussion, the authors are suggested to search database like Web of Science with searching title words ‘myofibril degradation’ AND ‘fish fillet’, to get related reference(s) for further discussion. 

The influence of fermentation process on the physiochemical properties, microbial diversity and volatile substance changes should be considered holistic way. They were not unrelated but closely related to each other. The science behind these changes is the food metabolomics. Therefore, to enhance the discussion, the authors are suggested to search database like Web of Science with searching title words ‘recent advances’ AND ‘metabolomics’ AND ‘food’, to get related reference(s) for further discussion.

In general, the language is acceptable. Minor revision is needed to go through the contents.

Author Response

(1) In this experiment, the control and blank were set up. Therefore, I think that the change in gel quality is mainly caused by fermentation, of course, it is very likely to be related to calcium ions. Therefore, the discussion on the influence of other factors has been added(ln325-326).

(2) There are few studies on the effect of microorganisms in Monascus on fish meat. The discussion of metabolites and microorganisms is increased(ln660-663,ln669-670, ln672-674).

(3) Other changes are indicated in blue fonts.

Round 2

Reviewer 1 Report

Please re-write abstract and conclusions. The abstract should be a brief summary of the significance of the research data presented towards potential industrial applications (purpose, approach, findings, important conclusions or questions). The conclusions should be summarize the main points and state the significance or results.

Please select an informative title, maybe “effects of Monascus-fermentation on the quality, microbial diversity, and volatile components of hairtail surimi”

 Please used LRI, linear retention index instead of CAS number.

 Legends of figure should be understand without reading manuscript, so it should include meaning of abbreviations. Please check legend of figure 8 (hairtail surimi??), also along the manuscript (ie.346, 335, 365, 683, 685, etc) since it is no applied to others species surimi.

 Figure 8 identify second line with English characters and trim imagen to enhance presentation.

 Please check spelling of Alaskan pollock should be “Alaska Pollock”

Ln 52: identify “TG” enzymes, ln 305 RDA: the first time of using abbreviation

 Re-write title of subheading of sections, for example, 3.2 “Results of determination…”. Maybe it will help if use three headings “1.Physicochemical properties, 2. microbial diversity and 3. Volatile and add in each one the subheadings of the corresponding determinations. Please follow the same structure in materials and cluster also in the results and discussion.

TPA is texture “profile” analysis

 Use international abbreviations for “sec”

 Fig4: please check units of WHC if it is in % should be around 60-80 %????

Please check “time/h” should be “time (h)”, WHC/ % should be “WHC (%)”

 Author Response

(1) The summary and conclusions were revised.

(2) The title changed to "Quality, microbial diversity and volatile components of Monascus-fermented hairtail surimi"

(3) Due to test reasons, the LRI value is not specifically obtained. So can CAS number be used instead of LRI?

(4) the legend of Figure 8 was revised, and the spelling of Alaskan pollock was corrected. “TG” enzymes and RDA were added with full writing.

(5) The subheading of sections "3 results and discussion" were re-write.

(6)  Other errors in the text have been modified.
